# Location of the Intermediate Echelon to Add Purchase Value and Sustainability Criteria in a Mining Supply Network

Rodrigo Barraza [1,2,*] , Juan M. Sepúlveda [1,*] and Ivan Derpich [1]

1   Department of Industrial Engineering, Universidad de Santiago de Chile, Santiago 9160000, Chile
2   School of Industrial Engineering, Universidad Santo Tomás, Santiago 8370003, Chile
*   Correspondence: rodrigo.barraza.a@usach.cl (R.B.); juan.sepulveda@usach.cl (J.M.S.)

**Abstract:** This study presents an operational analysis to determine the location of an intermediate hub in a supply network for the mining industry, incorporating sustainability criteria through an optimization model. The sector of small, medium, and artisan mining enterprises (PAMMA), in Chile, has the same pressure as large mining to meet the demands of sustainability in the medium term, but the network of PAMMA facilities is precarious and requires government support for development. One strategy to improve the supply network is to locate intermediate points with limited capacities (called purchasing powers) to help the viability of the business model by incorporating sustainability objectives, such as diminishing the movement of minerals, as well as reducing the carbon footprint and gas emissions, all in support the promotion of the activity of small miners. In order to achieve the strategy above, a mathematical model of location and sustainable capacity is proposed. A grouping of suppliers was carried out to establish the number of mining suppliers in each cluster and the location of the intermediate hubs. Then, the prioritization of the parameters and classification of the processing plant alternatives was performed to define a vector of weights to rank the degree of sustainability. A sustainability matrix was calculated on the basis of the distances and transportation costs between the supplier hubs of the supply network and the processing plants. With each of these factors, a capacity model was developed to validate the mineral process flows in the supply network and estimate the expected productivity levels. The model is intended to support operational decision making when determining the location of an intermediate purchasing power that reduces the impact of transportation costs and emissions. The model was applied in a case study of the supply network in the small mining sector in Chile. The results recommend the location of hubs to add value and encourage investment in the PAMMA supply network.

**Keywords:** sustainable operations; sustainable supply network; circular economy; circular supply chain



## 1. Introduction

This research for sustainable supply networks was carried out in the context of the primary sector of the economy, specifically, small, medium, and artisan mining enterprises (PAMMA), whose main purchasing power stems from state-owned metal refineries, i.e., the area that includes all economic activities dedicated to the extraction of natural resources, obtaining in exchange raw materials, which are earmarked for processing by the plants. This study seeks to identify the specific factors of sustainability for primary supply networks in the industry, particularly for mining, considering the impact of incorporating an intermediate step into the supply network, such as a mineral storage warehouse or deposit, in order to reduce transport costs in the operations of small miners and bring the purchasing power of the government or private sector buyers, closer to small producers.

Small and medium-sized mining is of great importance in Chile; proof of this is that there are several development institutions in the government that provide assistance to this sector, generating economic loans with preferential rates, as well as bringing the milling and purchase points to the sites distributed throughout the entire length of the country.

Further proof of this is that the government constantly promotes laws that benefit this sector. Former President Michelle Bachelet on 21 November 2016 initiated a bill to stabilize the price of copper for small-scale mining. As she quoted, "In Chile, small-scale mining has been present since before the founding of the Republic; since then, it has been a key economic activity in its development, both in the north and in the center of the country" [1]. In this context, cities and towns depend directly on this activity, generating employment and a productive chain. However, this sector is highly vulnerable to changes in metal prices in international markets; thus, the adoption of public policies that allow its stable development is justified. According to estimates by the Chilean Copper Commission (COCHILCO), the value of small-scale mining production in 2015 was approximately 320 million USD, corresponding to some 58,000 tons of fine copper. In terms of direct employment, assuming an average of seven people per job, small-scale mining comprises some 6300 workers. For its part, the average copper production of small-scale mining, between 2007 and 2015, was 81,000 fine tons. This is especially relevant in geographic areas where small-scale mining has a high impact, such as the Antofagasta, Atacama, Coquimbo, and Valparaíso regions in north of the country. In particular, it is the main economic support in those sectors characterized by less productive diversification and with operations far from the main centers of local consumption, according to data from the National Service of Geology and Mining (SERNAGEOMIN) for the year 2015. In this same sense, in accordance with the information of the National Mining Enterprise (ENAMI) regarding producers who make sales in some of its agencies, it is highlighted that small mining is an important source of work for micro-entrepreneurs and small entrepreneurs, with approximately 1312 producers registered by ENAMI, of which an average of 905 made regular deliveries in 2015. In addition, the sensitivity of small-scale mining associated with the price of copper stands out, since the number of producers with regular deliveries to ENAMI was, on average, 1566 in 2011, falling to the 905 mentioned above, due to the drop in the price of copper.

Currently, the business model consists of the delivery of mineral by the supplier of small mining and the purchase of minerals by state-owned metal refineries. Adding an intermediate step in the supply network contributes to sustainability, since the current configuration produces impacts or extra costs that threaten efficiency at the three levels of economic, environmental, and social impact [2]. According to the opinion of experts, seven criteria have been incorporated to establish the sustainability of the processing plants: productivity, efficient transportation, energy efficiency, infrastructure cost, water use, waste, and security.

In the last 10 years, global awareness has broadened to promote and accelerate the transition toward a circular economy that seeks to eliminate waste and help the regeneration of natural ecosystems. It has added to the previous trend of the impact of digital transformation and the incorporation of Industry 4.0 practices that create opportunities to manage the change of companies toward what is known as glocalization [3], i.e., business models with a global perspective, but whose behavior is local, with a focus on territorial management, making the regions or territories relate physically and virtually and creating a specific economic and social link for each community, but considering the impact or effects on a global scale.

In turn, this study seeks to carry out measurements and collect information to determine the level of efficiency in the sustainable supply network to support public investment management aimed at raising the standards of small-scale mining supplier networks. It is cheaper to design supply chain logistics with climate change impacts in mind from the start than to pay for disruptions and damages in the event of failures, considering the impact on local communities, conflicts, and operating costs and closing, since Chile joined the OECD and follows its indications about investments and sustainability [4]. This will make it possible to explain the behavior of an industrial sector and its evolution over time. From the problem posed above, the following research question arises: Where is the intermediate purchasing power for a sustainable mining supply network located?

Our research work contributes to the SSC literature and public policy development in several ways. In the first place, the model makes it possible to measure the costs ingroups or subgroups of companies that make up a supply network, incorporating in a novel way the sustainability index of the sector of companies that make up the network. Because global economies have entered an era of sustainable innovation distribution, the literature on SSC will benefit from a methodological proposal based on business networks such as the one presented in this research work. Secondly, the cost model based on company networks includes the positioning of companies in the primary sector of the economy in a competitive scenario when evaluating their levels of sustainability. Traditionally, these measures focus on the investments of each company, i.e., they treat companies as isolated entities. Thirdly, the research has been developed looking for a procedure that overcomes the persistent limitation that derives from the dependence on survey data and the unavailability of sustainability investment data for many companies. Only 2% of the companies in the region have registered to report their sustainability plans to the Ministry of the Environment. The sustainability index measurement approach has made it possible to base the findings on observations for 527 small-scale mining companies over a period of just over 3 years.

This article is organized as follows: Section 2 presents the literature review; Section 3 presents the research method and tools; Section 4 presents a case study; Section 5 presents the results; Section 6 presents the discussion. Conclusions and opportunities for future work are presented in Section 7.

## 2. Literature Review

A reference point in the literature is the sustainable supply chain (SSC), which, in general terms, can be defined as a system for the management of integrated operations of forward and reverse supply chains [5]. The SSC is an initiative that broadens the scope of value creation through product reconstruction activity [6]. As an important aspect of the circular economy and resource efficiency [7], the sustainable supply chain is seen as a promising strategy to ensure long-term availability of materials by creating additional sources of supply through process redesign and recycling [8].

Downsizing of the extractive industry and recycling reduce the need for minerals delivered by the mining industry and also reduce the negative environmental consequences of their extraction and processing [9]. The existing supply chain literature does not provide comprehensive indications for the development of SSCs. Current research rather focuses on SSC subsystems, such as reverse supply chain and reverse logistics, with limited understanding of the supply chain as a whole [10]. There is an imperfect overlap of the sustainable business model concept and its subcategories such as circular supply chain models [11].

In the industrial sectors of small producers, where there is little formalization of their activities, it reduces the efficiency of evaluations and the analysis of public policy proposals for solutions for the sector, while also making it difficult to apply promotion instruments, by both public and private institutions. PAMMA has its own dynamics and requires a particular analysis for each of the nodes in the supply network. It is necessary to consider the variations in the quantity produced, which is due to the intermittence of its producers that are sensitive to periods of activity defined by the cycle of metal prices, annual cycles, or seasonality [12]. Given the intermittency in production that impacts the logistics network with variations, it is necessary to incorporate the analysis of a sustainable supply where it is possible to visualize the limitations in the production, transport, and process capacity of the network system [13].

The circuits of the supply chain have development potential in the primary sector of the economies, as well as in the PAMMA, with an impact on climate change compared to other process redesign scenarios [14]. Currently, mining faces great challenges [15]. Staying competitive as an industry is critical [16], as the mining sector must deal with deteriorating grades in deposits, longer travel distances, higher hardness, and impurity

content in minerals, all normally associated with greater depth of deposits. For years, large mining companies have promoted the outsourcing of a significant number of supply and service functions, which generates an important development of external companies that form supply chains for the mining industry. This development has proven to be an effective mechanism to activate the dynamism of the mining sector toward a broader and more diverse set of sectors that shape the world economy [17]. Taking all this into account, supply chains play an essential role, both for the economy and for mining companies [18]. Therefore, it is necessary to have tools that allow managing this change, with the integration of public policies and operational plans of companies in the process of promoting the development of suppliers.

## 3. Methods and Tools

The method proposed to determine the location of an intermediate echelon to activate the purchasing power of the supply network is based on the phases shown in Figure 1 and seeks the development of small mining that can explain, predict, and understand the behavior of a sustainable supply chain.

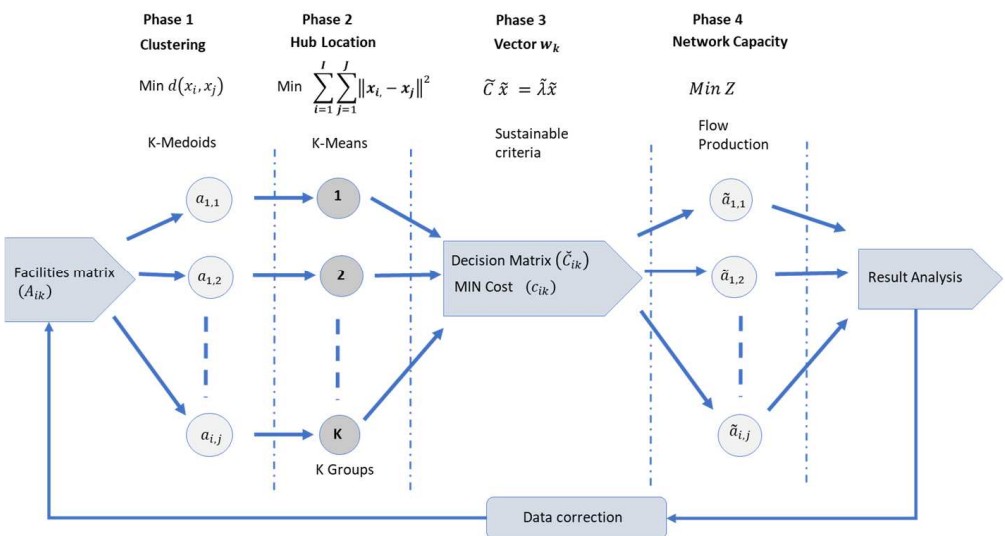

**Figure 1.** Phase diagram of the supply network optimization model.

### 3.1. Clustering Algorithm for Suppliers

According to phase 1 and phase 2 shown in Figure 1, the clustering algorithm allows $n$ nodes of the first step (extractive facilities) to be divided into $k$ groups; in order to perform this step, the *K*-Medoids algorithm groups the points of a matrix $X$ of observations, centered on the observation closest to the theoretical center. This is because *K*-Medoids seeks to split the observations. Next, the algorithm is described along with its cost function [19]. The cost function is based on the distance (no cost measure is modeled); the same cost is assumed as 1 USD for all distance measurements in kilometers, as defined by Equation (1):

$$Min\ f = \sum_{i=1}^{I}\sum_{j=1}^{J} d(x_i, x_j),\tag{1}$$

where $(x_i, x_j)$, corresponds to the location of each active small mining, $x_i$ is the latitude of point by active small mining $i$, $\forall i = 1, \ldots, I$, and $x_j$ is the longitude of point by active small mining $j$, $\forall j = 1, \ldots, J$.

The cost function seeks to minimize the computed Haversine distance [20] from each observation $(x_i, x_j)$ to the medoids.

1. The $k$ clusters are initialized, corresponding to the selection of $k$ random observations.

2. The Haversine distance from each observation to the potential $k$ medoid is measured by $d(x_i, x_j)$. Then, the observations that minimize the cost function are assigned to $z_j$ as defined in Equation (3). The symbol $\gamma$ is the radius of the Earth.

$$d(x_i, x_j) = 2rsin^{-1}\left(\sqrt{sin^2\left(\frac{x_{i+1} - x_i}{2}\right) + cos(x_i)cos(x_{i+1})sin^2\left(\frac{x_{j+1} - x_j}{2}\right)}\right) \quad (2)$$

$$z = \{x_{i,j}: \ d(x_i, x_j) \leq d(x_{i+1}, x_{j+1}) \ \forall i = 1, \dots, I; \ \forall j = 1, \dots, J\}. \quad (3)$$

3. The closest observation to the medoid is checked, and the current selection of medoids is changed.
4. If the cost function decreases from the previous selection, the process is iterated from step 2; otherwise, the algorithm ends.

*3.2. Sustainability Criteria*

According to phase 3 in Figure 1, an MCDM problem can be expressed as a decision matrix whose elements indicate the evaluation values of all alternatives $A_i$ with respect to each criterion $C_k$ under a fuzzy environment, and the ranking is obtained as follows:

1. We develop the decision matrix ($A$),

$$\begin{pmatrix} A_{11} & A_{12} \cdots & A_{1k} \\ A_{21} & A_{22} \cdots & A_{2k} \\ & \vdots & \vdots \\ A_{i1} & A_{i2} \cdots & A_{ik} \end{pmatrix}, \quad (4)$$

where $A_{ik}$ ($A_{ik} > 0$) denotes the performance value of the $i$ alternative on $k$ criterion, ($i = 1, \dots, n; \ k = 1, \dots, m$).

2. When calculating the entropy $e_k$ for each criterion, we can use

$$e_k = \left(\frac{-1}{ln\ m}\right)\left(\sum_{i=1}^{n} A_{ik}\ lnA_{ik}\right). \quad (5)$$

3. The criterion weight vector is

$$w_k = \frac{|1 - e_k|}{\sum_{k=1}^{m}|1 - e_k|} \quad (6)$$

4. Max–min normalization is performed for each decision maker, and the weighted normalized decision matrix is calculated

$$\hat{A}_{ik} = \begin{cases} \dfrac{A_{ik}}{Max_i\{A_{ik}\}} & if\ k\ benefit \\ \dfrac{A_{ik}}{Min_i\{A_{ik}\}} & if\ k\ cost \end{cases}. \quad (7)$$

$$a_{ik} = w_j\hat{A}_{ik}. \quad (8)$$

5. The negative ideal solution points are determined.

$$|Min(a_{ik})|_{1\times m}. \quad (9)$$

6. The Euclidean and absolute distances are calculated.

$$Euc(a_{ik}, Mina_{ik}) = \sqrt{\sum_{i=1}^{I}(a_{ik} - Min(a_{ik}))^2}. \quad (10)$$

$$Abs(a_{ik}, Mina_{ik}) = \sum_{i=1}^{I} |a_{ik} - Min(a_{ik})|. \tag{11}$$

7. The relative assessment matrix is constructed and the assessment score and rank are calculated.

$$RA = |s_{ij}|_{n \times m}. \tag{12}$$

$$s_{ij} = (Euc_i - Euc_j) + (\alpha(Euc_i - Euc_j)(Abs_i - Abs_j)). \tag{13}$$

The decision maker can set the parameter in this study ($\alpha$ = 0.02).

8. The assessment score and rank are calculated. The largest value yields the best alternative.

$$AS_i = \sum_{j=1}^{J} s_{ij}. \tag{14}$$

### 3.3. Network Capacity

According to phase 4 in Figure 1, the total cost optimization problem has the capacity restrictions of the supply network, the restriction of the maximum production capacity of the processing plants, the ore supply capacity from the mining sites, and the operating costs.

#### 3.3.1. Definitions

The indices of each of the network nodes are represented as follows:

- $v = \{1, \ldots, V\}$ corresponds to the set of active small mining operations;
- $h = \{1, \ldots, H\}$ corresponds to the clusters;
- $i = \{1, \ldots, I\}$ corresponds to the set of processing plants;
- $t = \{1, \ldots, T\}$ corresponds to the evaluation periods.

#### 3.3.2. Variables

The variables for the network capacity problem function are represented as follows:

- $Z$ is the operation result;
- $q_{vh}$ is the amount product from mine site $v$ to hub $h$;
- $m_{ih}$ is the amount product from hub $h$ to processing plant $i$;
- $b1_{ht}$ is the product stock in the hubs.

#### 3.3.3. Parameters

The configuration parameters of the network capacity problem are represented as follows:

- $p_{vh}$ is the production cost in small mining operations;
- $Q_h$ is the maximum capacity of hub $h$;
- $o_{ih}$ is the production cost for processing plant $i$;
- $M_i$ is the maximum capacity of processing plant $i$;
- $d_{ih}$ is the distance between the $H$ hubs operations and $i$ processing plants;
- $AS_i$ is the sustainable assessment score for processing plants;
- $L_{vh}$ is the percentage of product losses in hubs;
- $T_i$ is the transportation cost based on diesel oil [\$/ton-km];
- $E_i$ is the cost of emissions that impact the environment.

#### 3.3.4. Constraints

The constraints for the network capacity problem function are represented as follows:

$$\sum_{v \in V} q_{vh} \leq Q_h \qquad \forall\, h \text{ hub capacity.} \tag{15}$$

$$\sum_{h \in H} m_{ih} \leq M_i \qquad \forall\, i \text{ processing plant capacity.} \tag{16}$$

$$b1_{h(t-1)} + \sum_{v \in V} q_{vh}(1 - L_{vh}) = b1_{ht} + \sum_{i \in I} m_{ih} \qquad \forall\, h, t \text{ product stock in the hub.} \tag{17}$$

$$q_{vh}, m_{ih}, b1_{ht} \geq 0 \qquad \forall\, h, t. \tag{18}$$

Moreover, following condition must hold:

$$\sum_{v \in V} s_v \geq 0 \qquad \text{number of suppliers.} \qquad (19)$$

### 3.3.5. Objective Function

The objective function seeks the minimum total cost and is represented as follows:

$$MinZ = \sum_{v \in V} \sum_{h \in H} p_{vh} q_{vh}(1 - L_{vh}) + \sum_{i \in I} \sum_{h \in H} o_{ih} m_{ih}(1 - AS_i) + \sum_{i \in I} \sum_{h \in H} d_{ih} m_{ih}(T_i + E_i) \qquad (20)$$

## 4. Case Study

PAMMA is the acronym and concept that brings together small, medium, and artisanal mining companies that exploit deposits and benefit directly from their mineral production. It is an activity with a high social impact. In the case of Andean countries, small-scale mining is a form of work, as well as a way of life rooted in cultural traditions. In the case of Chile, PAMMA generates around 50,000 direct and indirect jobs [21]. From the point of view of production in Chile, we can find an approximate number of 2000 deposits associated with PAMMA, which are estimated to create some 35,000 direct jobs in the country. This PAMMA group produces around 5% of the copper at the national level. However, the contribution to the economy by PAMMA is not visible because the productive and exporting role is included jointly with the indicators of large-scale mining.

### 4.1. Case Setting

This study considers a deepening in the small mining operation for five relevant zones in Chile. At this stage, the research has a special focus on the mineral supply processes by suppliers of the PAMMA sector to the operations of the mineral processing plants in Chile. Table 1 shows the processing plants.

**Table 1.** Alternative processing plants.

| Alternatives | Zone | Plant | Capacity (Tons/Day) |
|---|---|---|---|
| A1 | Coquimbo | Delta | 1667 |
| A2 | Atacama | Matta | 3080 |
| A3 | Atacama | Vallenar | 694 |
| A4 | Atacama | El Salado | 1495 |
| A5 | Antofagasta | Taltal | 940 |

### 4.1.1. Clustering for Mining Suppliers

The parameters used in the classification of the facilities of mining suppliers are observed in Table 2. The number of mining facilities corresponds to the set of active small mining operations. The other parameters, such as the number of criteria, decision variables, weight of the criteria, and the number of clusters are entered in the clustering algorithm.

**Table 2.** Ranking parameters for small mining.

| Parameters | Value |
|---|---|
| Number of mining facilities | 527 |
| Number of criteria | 3 |
| Decision variables | Volume; distance; height/slope |
| Weight of criteria | 1/3 |
| Number of clusters | 4 |

The location of the hubs is shown in Figure 2, and their corresponding points are shown in Table 3. The proposed methodology of combined algorithms to optimize transport presents the best behavior with four hubs for mineral transfer from small mining operations.

In the next step, the optimal capacity for the flow of ore from the hubs to the processing plants is defined.

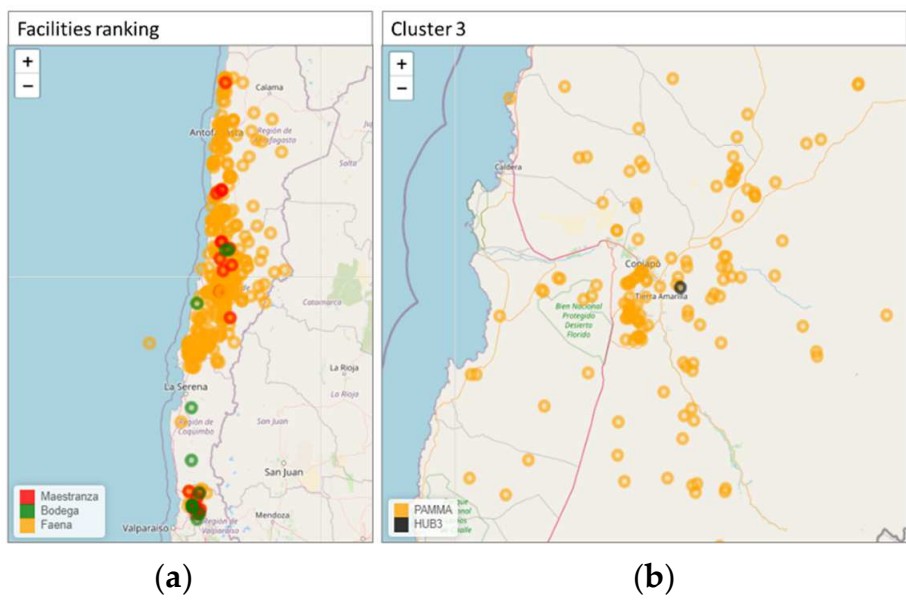

**Figure 2.** Diagram of the solutions of the optimization algorithm model: (**a**) description of facilities ranking; (**b**) description of *k*-means cluster number 3.

**Table 3.** Locations obtained by the optimization algorithm, n = 4.

| Hubs | Datum (Long/Lat) | Number Suppliers |
|---|---|---|
| Hub 1 | −70.425; −28.497 | 147 |
| Hub 2 | −70.581; −26.375 | 110 |
| Hub 3 | −70.198; −27.440 | 145 |
| Hub 4 | −70.239; −24.852 | 125 |

4.1.2. Sustainability Criteria for Plants Process

Once the location of the hubs for each of the supplier clusters has been defined, we establish the vector of weights for the processing plants according to the sustainability criteria provided by the supply experts, and we can identify the score of each of the plants of the supply network, as detailed below. The evaluation was performed according to the methodology of collecting the preferences of decision makers regarding five alternatives for processing plants in Chile (Table 1) and the seven sustainability criteria shown in Table 4.

**Table 4.** Sustainability criteria.

| Criteria | Description—Optimization |
|---|---|
| C1 | Productivity—Max |
| C2 | Efficient transportation—Max |
| C3 | Energy efficiency—Max |
| C4 | Infrastructure cost—Min |
| C5 | Water Use—Min |
| C6 | Waste—Min |
| C7 | Security—Max |

Table 5 shows the result of the experts' qualifications for each of the sustainability criteria in each processing plant. Evaluations are based on scores for each criterion. The optimization of the criteria C1, C2, C3, and C7 is maximization. The optimization of the criteria C4, C5, and C6 is minimization.

Applying Equations (5) and (6), we obtain that the vector of weights used is $w_k$ = (0.131, 0.134, 0.125, 0.130, 0.143, 0.168, 0.168), where the greatest impact on the weight for decision makers is found in the criteria of variations in management waste and industrial security. The decision matrix $A_{ik}$ in Table 6 represents the aggregate preference of the decision makers with respect to the criterion $C_k$ over the option $A_i$.

**Table 5.** Expert rating by criteria.

| Alternatives | C1 | C2 | C3 | C4 | C5 | C6 | C7 |
|---|---|---|---|---|---|---|---|
| A1 | 105 | 65 | 8 | 80 | 67 | 10 | 5 |
| A2 | 78 | 115 | 9 | 70 | 45 | 50 | 4 |
| A3 | 74 | 85 | 7 | 60 | 33 | 40 | 4 |
| A4 | 63 | 90 | 8.5 | 50 | 44 | 30 | 1 |
| A5 | 62 | 60 | 7.5 | 50 | 26 | 20 | 2 |

**Table 6.** Weighted normalized decision matrix.

| Alternatives | C1 | C2 | C3 | C4 | C5 | C6 | C7 |
|---|---|---|---|---|---|---|---|
| A1 | 0.131 | 0.076 | 0.111 | 0.130 | 0.055 | 0.168 | 0.034 |
| A2 | 0.097 | 0.134 | 0.125 | 0.114 | 0.082 | 0.034 | 0.042 |
| A3 | 0.092 | 0.099 | 0.098 | 0.098 | 0.112 | 0.042 | 0.042 |
| A4 | 0.079 | 0.105 | 0.118 | 0.081 | 0.084 | 0.056 | 0.168 |
| A5 | 0.077 | 0.070 | 0.105 | 0.081 | 0.143 | 0.084 | 0.084 |

Table 7 shows the results assessment score by processing plants, where the best evaluation for sustainability is for the A4 processing plant, located in El Salado. On the other hand, the worst evaluation regarding sustainability criteria is for the A3 processing plant, located in Vallenar.

**Table 7.** Score by processing plants.

| Alternative | Score |
|---|---|
| A1—Delta | 0.155 |
| A2—Matta | −0.178 |
| A3—Vallenar | −0.212 |
| A4—El Salado | 0.193 |
| A5—Taltal | 0.043 |

## 5. Results

In this section, we calculate the capacity optimization, and then use the solutions of the classification algorithms, the location algorithm, and the evaluation ranking of the sustainability criteria. Table 1 shows the processing capacity of each of the processing plants of the supply network.

### 5.1. Network Capacity

Since our objective is to establish the capacity of the network, for the purposes of this phase of the study, we assume the cost of transportation with a value of 1 USD per kilometer. The results for simulation scenarios are shown below. The first scenario is without hubs, i.e., without intermediate points. The second scenario considers the hubs in the supply network.

#### 5.1.1. Hubless Scenario

Figure 3 shows the operation flowchart in the supply network without an intermediate node to accumulate stock. In this simulation scenario, the total cost of the supply network is 2,500,500 USD, which represents the worst-case scenario for the set of active small mining operations.

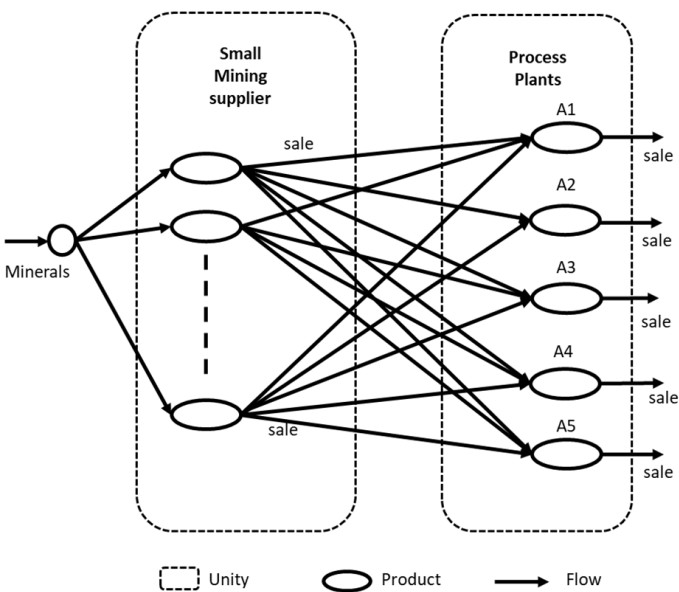

**Figure 3.** Operation flowchart.

### 5.1.2. Scenario with Intermediate Hubs

In this simulation scenario, the total cost of the supply network is 2,929,616 USD, where less than 20% of the operating costs of the supply network are explained by the production and dispatches from the small mining suppliers to the hubs, and more than 80% of the costs of the network are explained by stock management in the hubs and dispatch from the hubs to the processing plants and the production cost in the plants. Table 8 shows the dispatch costs from the hubs to the processing plants. Figure 4 shows the results of the operation flowchart when adding an intermediate step in the supply network in PAMMA.

**Table 8.** Dispatch cost (USD).

| Alt. | Hub1 | Hub2 | Hub3 | Hub4 |
|------|------|------|------|------|
| A1 | 950 | 1173 | 1278 | 587 |
| A2 | 816 | 774 | 51 | 1923 |
| A3 | 264 | 1681 | 952 | 1828 |
| A4 | 935 | 156 | 391 | 938 |
| A5 | 1849 | 644 | 1267 | 335 |

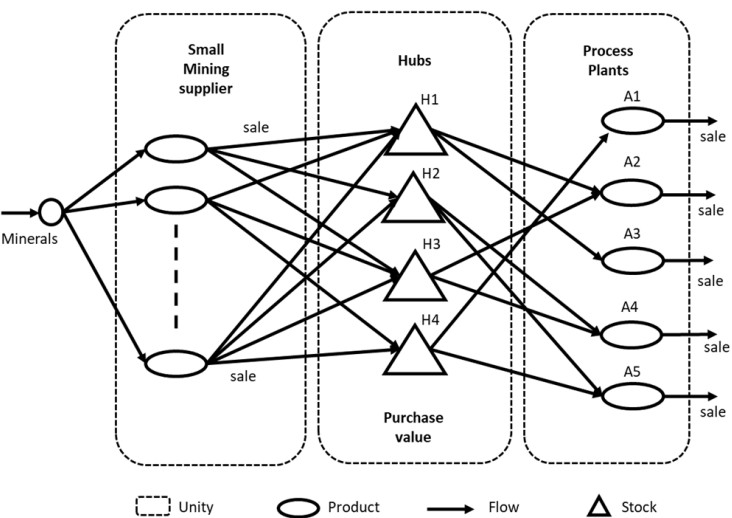

**Figure 4.** Flowchart of the proposed operation.

Table 9 shows the maximum production capacity for each of the PAMMA provider nodes.

**Table 9.** Maximum transfer capacity from hubs to processing plants.

| Cluster and Hub | Capacity (Tons/Day) |
|---|---|
| Cluster—H1 | 2072 |
| Cluster—H2 | 1565 |
| Cluster—H3 | 2044 |
| Cluster—H4 | 1762 |
| Total | 7876 |

The results for the processing capacity of the supply network are shown in Table 10.

**Table 10.** Supply network ore processing capacity, in TMF per day.

| Processing Plants | Cluster—H1 | Cluster—H2 | Cluster—H3 | Cluster—H4 |
|---|---|---|---|---|
| A1—Delta | 342 | 0 | 0 | 892 |
| A2—Matta | 1036 | 0 | 2044 | 0 |
| A3—Vallenar | 694 | 0 | 0 | 0 |
| A4—El Salado | 0 | 1495 | 342 | 0 |
| A5—Taltal | 0 | 70 | 0 | 870 |

## 6. Discussion

The sustainable business model in a supply network is the organization of its activities seeking to optimize its competitive position. This includes defining what and how much product to buy and from whom. In the PAMMA mining sector, the supply network faces the dilemma of favoring its operation upstream or downstream in the value chain [22]. When applying the model, the results of the simulation of scenarios recommend the location of four hubs, intermediate nodes, to add value and encourage investment in the PAMMA supply network. By adding the intermediate steps in the supply chain, total costs increase by more than 16%. However, more than 80% of transportation and production costs are transferred from the small miners to the processing plants.

We observe when comparing the scenarios that the supply networks and their companies that remain with the same technological standard without incorporating environmental stewardship practices or technology have a lower increase in their operating costs. On the other hand, companies that make investments and technological changes that seek energy efficiency and sustainable practices in their operations and transportation that contribute to caring for the environment have higher initial operating costs. The advantage of energy efficiency technologies for the networks of companies that make the investments is that the variable costs are lower; for this reason, the decrease in operating costs can only be appreciated when the mining companies that make the technological changes have a higher production volume. An interesting aspect to deepen is to incorporate sustainability in the supply network where a distributive effect occurs for the companies that comprise it. The average income of the mining supply network that does not apply sustainability and energy efficiency standards can grow. This reflects that the companies that made changes had a higher average cost of operating their processes. Figure 5 shows the results of the location algorithm for the processing plants in yellow and the location proposed by the location algorithm for the purchasing power hubs in black.

One observation derived from this research is that, given the scenario of environmental contingency due to climate change, mining companies should implement circular economy and energy efficiency technologies, which are supposed to be cheaper. Could the results change? According to the results obtained, we cannot yet say that the costs of the supply chain would fall with the introduction of energy efficiency technologies and environmental practices in companies over supply networks that do not make improvements in their processes. On the other hand, if the cost estimator had shown an increase in the value of an order of magnitude similar to the variation in the sustainability score by processing plants,

it could not be said that there would be an increase in costs either. A measurement error would be enough to reject the following hypotheses:

- Sustainability increases supply network costs;
- Sustainability reduces supply network costs.

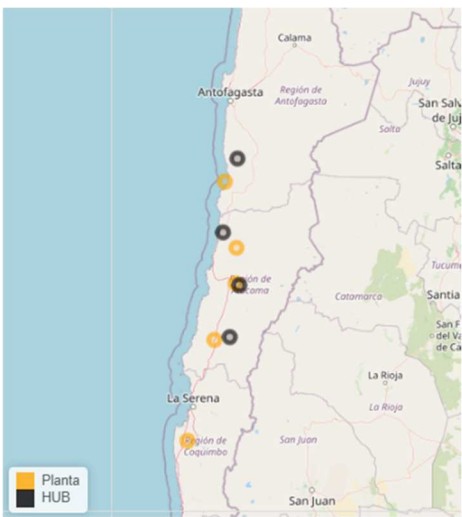

**Figure 5.** Diagram for the purchasing power small mining hubs.

It is still a matter of debate whether investments in sustainable practices and energy-efficient technologies have a positive impact on operating costs in the network. However, if the data existed in the PAMMA sector, the calculation would have been performed in the same way: the operating costs of a sustainable supply network would be calculated, the costs of a network without sustainable practices would be introduced, the cost of inputs would be calculated, and the cost change for the supply network would be calculated. However, when reviewing the two previous hypotheses, considering the economies of scale, the two hypotheses can be accepted. In the case of companies that do not make technological changes, their costs would be low. In this case, the result of the change would depend on the loss of economies of scale and the cost levels of the companies that make up the PAMMA supply network.

There is a permanent effort by small mining operations to lower their production costs, ensuring an improvement in their global value given the price at which the government agency will make payments for minerals. The challenge is to show the commitment to remain in the market, making investments in technologies that are friendly to the environment and contribute to reducing the carbon footprint. These investments are irreversible and tend to increase fixed costs and decrease variable costs, thereby increasing the exit barriers of a company in the PAMMA sector.

## 7. Conclusions and Future Research

In this article, we presented a method for the redesign of the supply chain of the small, medium, and artisanal extractive mining sector in order to minimize the costs of transportation between these locations and the plants which process the mineral extracted at those facilities. Since the extractive sites are numerous and spread over an extended geographical region, the proposal is based on identifying centroids or medoids to become hubs or concentration sites to gather the production to be sent to the processing plants. That is, instead of a direct transportation mode, a hub is determined to receive and send the material to the next step of the supply chain. This configuration is expected to contribute to sustainability by minimizing transportation costs due to economies of scale in the movement of a larger amount of material and reducing the $CO_2$ emissions.

The approach to locate the hubs was the *K*-medoids method followed by a refined search by *k*-means. The numerous mining operations were grouped by the *K*-medoids

algorithm by identifying clusters of working sites; for each cluster, a centroid was found such that the distances within the cluster were minimized. Once the source nodes of the clusters were found, a transport optimization model was developed in order to find the flows toward the plants that minimize the total cost. By minimizing costs, the global value of the chain was maximized.

A case study in the northern region of Chile was presented showing the sequential steps of the models. Ongoing research is addressing other issues to further investigate the effect of sustainable technologies over the optimization of the supply chain operations, such as renewable energy sources, cleaner transportation, and circular economy practices.

**Author Contributions:** Conceptualization, R.B.; methodology, R.B. and J.M.S.; testing of formulation, R.B., J.M.S. and I.D.; software, R.B.; writing and editing, R.B., J.M.S. and I.D. All authors have read and agreed to the published version of the manuscript.

**Funding:** This research was funded by DICYT-USACH, Grant N°061119SS, Grant N°062117DC and The Industrial Engineering Dept. USACH.

**Institutional Review Board Statement:** Not applicable.

**Informed Consent Statement:** Not applicable.

**Data Availability Statement:** Not applicable.

**Acknowledgments:** The authors gratefully acknowledge the support of CIGOM and the Sustainable Development Division, Ministry of Mining, Government of Chile.

**Conflicts of Interest:** The authors declare no conflict of interest.

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
