# Peer review of "Location of the Intermediate Echelon to Add Purchase Value and Sustainability Criteria in a Mining Supply Network"

_sustainability, doi:10.3390/su141912920_

Round 1
Reviewer 1 Report
A mathematical model of location and sustainable capacity is proposed in the investigation. The structure of the manuscript needs to be improved. The current manuscript could not meet the creterion of the journal. Some comments are provided below:
1. The abstract should be mainly focus on what this paper has done, too many background information should be avoided.
2. The authors should show how their model bahaves, and is it good in the abstract.
3. The introduction is only with 1 paragraph, it is suggested to be divided into several parts.
4. 'Productivity; Effcient transportation; Energy effciency; Infrastructure; Water Use; Waste; and Security.', why 'infrastructure' is standing alone? Is it 'infrastructure cost'?
5. 'Motivation and Objectives' has too long a paragraph, it is suggested to divide them into several parts.
6. 'Where xij corresponds to the location of each of active small mining;
Where yij corresponds to the location of processing plants', check in other literature how to define parameters in the equations.
7. Figure 5 is with low picture quality.
8. 'There is a permanent effort by small ', spaces should be added before.
9. The conclusion is also with justa single paragraph, which should be divided.
10. Where is the cost function (Equation 2) come from? What is the unit for x, and y?
11. All the parameters in the equations should be defined, but the struructure of the paper should be reorganized, please check the other papaers for parameters definition.
12. Strictly, all the equations should have reference literature.
13. It seems that the destination has not been determined, if you just calculate the optimization method, without destination and other parameters, is this investigation close to reality?
14. What is new about this paper?
Author Response
Point 1: The abstract should be mainly focus on what this paper has done, too many background information should be avoided.
Response 1: We appreciate your suggestions. We have made changes in the abstract, the text of line 6 was eliminated and a text is added in line 20 about the results.
Point 2: The authors should show how their model behaves and is it good in the abstract.
Response 2: We appreciate your comment. We have made changes in the abstract, the behavior of the model between lines 17 to 21 is explained.
Point 3: The introduction is only with 1 paragraph; it is suggested to be divided into several parts.
Response 3: We appreciate the suggestion. Section 1, Introduction, was divided into 7 paragraphs according to the topics, at lines 25 to 114.
Point 4: 'Productivity; Efficient transportation; Energy efficiency; Infrastructure; Water Use; Waste; and Security.', why 'infrastructure' is standing alone? Is it 'infrastructure cost'?
Response 4: Indeed, the correction is made to "infrastructure cost", at lines 73 to 253.
Point 5: Motivation and Objectives' has too long a paragraph, it is suggested to divide them into several parts.
Response 5: We appreciate the suggestion. The text was divided into 3 paragraphs and it was moved to the introduction, in line 94.
Point 6: Where xij corresponds to the location of each of active small mining; Where yij corresponds to the location of processing plants', check in other literature how to define parameters in the equations.
Response 6: Indeed, the correction is made to only corresponds to the location of each active small mining in phase 1 and phase 2, in line 170.
Point 7: Figure 5 is with low picture quality.
Response 7: Indeed, the correction is made to improve picture quality in Figure 5 and Figure 2. RStudio Version 1.2.1335 open-source app has limitations on the number of pixels in the output of the flexdashboard function.
Point 8: There is a permanent effort by small ', spaces should be added before.
Response 8: Indeed, the correction is made on line 342 of the manuscript.
Point 9: The conclusion is also with just a single paragraph, which should be divided.
Response 9: We appreciate the suggestion. Section 7, Conclusions, was divided into 3 paragraphs, according to the idea, at lines 361 to 371.
Point 10: Where is the cost function (Equation 2) come from? What is the unit for x, and y?
Response 10: Suggestion is added in the first paragraph of subsection 3.1 at line 168.
Point 11: All the parameters in the equations should be defined, but the structure of the paper should be reorganized, please check the other papers for parameters definition.
Response 11: We appreciate the suggestion. We reorganized the definition of the parameters for Section 3, at lines 157 to 219.
Point 12: Strictly, all the equations should have reference literature.
Response 12: We appreciate the suggestion. We have checked the references to the literature.
Point 13: It seems that the destination has not been determined, if you just calculate the optimization method, without destination and other parameters, is this investigation close to reality?
Response 13: We appreciate the comment. Confirm, the data used corresponds to the actual location of the facilities and is a contribution to support the development of public policies.
Point 14: What is new about this paper?
Response 14: We appreciate the comment. Suggestion is added in the sixth paragraph of Section 1, Introduction, at lines 94 to 110.

Reviewer 2 Report
My comments are as below:
1. The introduction of the paper can be improved. Please make it in a multiple paragraphs instead in one long paragraph that made difficult for reader to capture what is this paper all about.
2. Figure 1 should be located in Section 3 and not under introduction section.
3. Check for symbol whether the letter should be italic or not. For example: in line 111, k groups and at Table 3: Locations obtained by the optimization algorithm n=4.
4. Tab on the last paragraph at line 270.
5. 5.2. Sustainable business model can be made into two paragraphs.
Author Response
Point 1: The introduction of the paper can be improved. Please make it in a multiple paragraphs instead in one long paragraph that made difficult for reader to capture what is this paper all about.
Response 1: We appreciate the suggestion. Section 1, Introduction, was divided into 7 paragraphs.
Point 2: Figure 1 should be located in Section 3 and not under introduction section.
Response 2: Indeed, the correction is made by moving Figure 1 to Section 3.
Point 3: Check for symbol whether the letter should be italic or not. For example: in line 111, k groups and at Table 3: Locations obtained by the optimization algorithm n=4.
Response 3: We appreciate the suggestion. Change to italic letter "n" in the title for table 3. The correction is under line 245 of the manuscript.
Point 4: Tab on the last paragraph at line 270.
Response 4: Indeed, the correction is made on line 342 of the manuscript.
Point 5: 5.2. Sustainable business model can be made into two paragraphs.
Response 5: We appreciate the suggestion. Section 6, Discussion, was added. The correction is made at lines 296 and 305 of the manuscript.

Reviewer 3 Report
In order to improve the quality of this study, some comments have been given as below.
1). In section 1, there are only one paragraph. We suggest to combine section 1 and section 2 into new introduction section.
2).In introduction, please highlight the motivation of this work, and authors should provide the academic evidence to support their viewpoints.
3).In introduction section, please provide some statistics or government published numbers to clarify the importance of studied problem.
4).In methodology section, please follow Figure 1 to illustrate the implement procedure phase by phase.
5).Subsections 3.3.1~3.3.4 only contains equations or notations. Please add some statements to describe them. It's easy for readers to follow.
6).A literature review section should be included. Then, we can know the development of related works.
7).One paragraph at least should contain thre sentences. Please check the 1st paragraph of subsection 4.1, the 1st and 2nd paragraph of 4.1.1, the 2nd and 4th paragraph of 4.1.2, the 1st paragraph of 5.1,the 1st and 2nd paragraph of 5.1.1, and so on.
8). In seciton 5, the discussion section should be separate into an independent section.
9)Conclusion section should be enhanced.
Author Response
Point 1: n section 1, there are only one paragraph. We suggest to combine section 1 and section 2 into new introduction section.
Response 1: We appreciate your suggestion. Section 1, Introduction, was reorganized and divided into 7 paragraphs, according to the topics, at lines 25 to 114.
Point 2: In introduction, please highlight the motivation of this work, and authors should provide the academic evidence to support their viewpoints.
Response 2: We appreciate the comment. Suggestion is added in the sixth paragraph of Section 1, Introduction, at lines 94 to 110.
Point 3: In introduction section, please provide some statistics or government published numbers to clarify the importance of studied problem.
Response 3: We appreciate the comment. Suggestion is added in the second paragraph of Section 1, Introduction, at lines 36 to 66.
Point 4: In methodology section, please follow Figure 1 to illustrate the implement procedure phase by phase.
Response 4: We appreciate the suggestion. Section 3 was reorganized, in each subsection the procedure phase is indicated, at lines 157 to 219.
Point 5: Subsections 3.3.1~3.3.4 only contains equations or notations. Please add some statements to describe them. It's easy for readers to follow.
Response 5: We appreciate the suggestion. Section 3 was reorganized, in each subsection was added explanations to describe them, at lines 157 to 219.
Point 6: A literature review section should be included. Then, we can know the development of related works.
Response 6: We appreciate the comment. Section 2, Literature Review, was added in line 116.
Point 7: One paragraph at least should contain thre sentences. Please check the 1st paragraph of subsection 4.1, the 1st and 2ndparagraph of 4.1.1, the 2nd and 4th paragraph of 4.1.2, the 1stparagraph of 5.1, the 1st and 2nd paragraph of 5.1.1, and so on.
Response 7: We appreciate the comment. Suggestion is added in the first paragraph of subsection 4.1 in line 233; Suggestion is added in the first and second paragraph of subsubsection 4.1.1 at lines 237 and 241; Suggestion is added in the second and fourth paragraph of subsubsection 4.1.2 at lines 254 and 263; Suggestion is added in the first paragraph of subsection 5.1 in line 274; Suggestion is added in the first paragraph of subsubsection 5.1.1 in lines 280.
Point 8: In seciton 5, the discussion section should be separate into an independent section.
Response 8: We appreciate the comment. Section 6, Discussion, was added in line 295.
Point 9: Conclusion section should be enhanced.
Response 9: We appreciate the suggestion. Section 7, Conclusions, was divided into 3 paragraphs according to the idea, at lines 361 to 371.

Round 2
Reviewer 1 Report
The paper has improved a lot, no comments now.
Reviewer 3 Report
All of my comments have been fully considered in this revised version.